# Development of Water Level Prediction Models Using Machine Learning in Wetlands: A Case Study of Upo Wetland in South Korea

**Changhyun Choi [1], Jungwook Kim [1], Heechan Han [2], Daegun Han [3] and Hung Soo Kim [3,*]**

[1] Institute of Water Resources System, Inha University, Michuhol-Gu, Incheon 22212, Korea;
    karesma0cch@naver.com (C.C.); rlawjddnr1023@gmail.com (J.K.)
[2] Department of Civil and Environmental Engineering, Colorado State University, Fort Collins,
    CO 80523, USA; heechan.han@colostate.edu
[3] Department of Civil Engineering, Inha University, Michuhol-Gu, Incheon 22212, Korea; eorjs0615@naver.com
* Correspondence: sookim@inha.ac.kr; Tel.: +82-32-874-0069

**Abstract:** Wetlands play a vital role in hydrologic and ecologic communities. Since there are few studies conducted for wetland water level prediction due to the unavailability of data, this study developed a water level prediction model using various machine learning models such as artificial neural network (ANN), decision tree (DT), random forest (RF), and support vector machine (SVM). The Upo wetland, which is the largest inland wetland in South Korea, was selected as the study area. The daily water level gauge data from 2009 to 2015 were used as dependent variables, while the meteorological data and upstream water level gauge data were used as independent variables. Predictive performance evaluation using RF as the final model revealed 0.96 value for correlation coefficient (CC), 0.92 for Nash–Sutcliffe efficiency (NSE), 0.09 for root mean square error (RMSE), and 0.19 for persistence index (PI). The results indicate that the water level of the Upo wetland was well predicted, showing superior results compared to that of the ANN, which was used in a previous study. The results intend to provide basic data for development of a wetland management method, using water levels of previously ungauged areas.

**Keywords:** machine learning; ungauged area; Upo wetland; water level; wetland

## 1. Introduction

Wetlands play important environmental, ecological, and hydrological roles, such as maintaining species diversity, providing nutrients, controlling water levels, and reducing floods and droughts [1]. Hydraulic and hydrological conditions, such as flow rate, water level, and inundation depth, are significant constraints for aquatic environments, including wetlands [2,3]. In particular, wetland species have their preferred inundation depths or permissible water level limits. Therefore, water level measurement and prediction are known to be important for proper wetland management and protection [4–7]. Environmental engineers and scientists are well-aware of the importance of measuring and predicting water levels in wetlands [8]. However, water level measurement is limited, especially in some wetland regions. In South Korea, a concrete example of this is the inadequately monitored Ramsar Designated Wetlands. This inadequacy in monitoring is due to the low budgets given for wetland research. Also, researchers and environmental engineers measure the water level only if necessary, but when the study is over, they stop conducting further monitoring. This is why it is difficult to secure wetland data in the long term [9].

Water level estimation is usually done using a process-based model. These models should be carefully selected to predict actual water level changes. A large number of parameters, such as

influencing factors that affect the water level, require much time to calibrate so as to secure the accuracy of the model prediction [10]. Because process-based methodologies take too long to model [11], recent studies have predicted the water level using a data-based model of machine learning (ML), such as an artificial neural network (ANN) [12]. In the early days, this method was widely used for real-time water level and flood forecasting, due to its fast modeling time [13–17]. Since then, ML has increasingly been used alone or in combination with other process-based models. ML, such as ANNs, have been widely used to predict water levels [18]. In fact, since the 1990s, water level prediction has been conducted to improve accuracy by applying various neural network modeling techniques [19–24].

The main objective of ML is to predict the dependent variables for a new independent variable based on the learned relationship between the two. Here, ML can be classified into supervised learning and unsupervised learning depending on the presence or absence of the dependent variables [25,26]. Representative supervised learning methods for classification and regression include the decision tree (DT) [27], random forest (RF) [28], and support vector machine (SVM) [29] methods. K-means for clustering [30] and self-organizing maps [31] are representative techniques for unsupervised learning. The advantage of ML is that various techniques can be applied depending on the user's application purpose. Each technique has also shown good performance, thus ML can complement the limitations of physical processes based on complex theories and mathematical equations.

In recent times, it has become possible to take into account all the factors affecting water level fluctuation, due to the development of computer technology for processing big data. ANN is widely used to forecast the groundwater level [32], water level in rivers [33–35], reservoirs [36–38], and wetlands [39,40]. However, research into water level prediction in wetlands is more recent than other areas of study, with the research focusing on the use of ANN [39,40]. Unlike studies on water level prediction in rivers, reservoirs, and groundwater, where ML is applied to evaluate and compare local ML models, studies using ANN alone could not be used to compare and evaluate the applicability to various ML models. In particular, even though collecting the water level data of wetlands is important for sustainable use of the wetland and the protection of ecosystems, studies on water level prediction in wetlands lag far behind those on rivers, groundwater and reservoirs, owing to a lack of data for wetlands necessary to establish models. Therefore, this study aims to develop a model of water level prediction suitable for wetlands using various ML models, such as DT, RF, and SVM, along with the existing ANN.

This study selected the Upo wetland, the largest inland Ramsar wetland in Korea, as the focus of our study. The water level data of the Upo wetland were used as a dependent variable while the weather data from the weather station closest to the Upo wetland and the water level data from the Mokpo embankment and Sindang Drainage Pump Station located upstream of Upo wetland were set up as the independent variables. To prevent over-fitting of the model, the data were divided into training and test dataset. Using the data of the training section and various statistical models such as the DT, RF, and SVM, as well as ANN, which were mainly used in previous studies, a model for water level prediction in the Upo wetland model was developed. The predicted water level was calculated by inputting the developed model into the data of the test section, and the prediction performance by each model was evaluated using evaluation indicators, such as correlation coefficient (CC), Nash–Sutcliffe efficiency (NSE), root mean square error (RMSE), peak value, and persistence index (PI). Figure 1 shows the flow chart of the overall study.

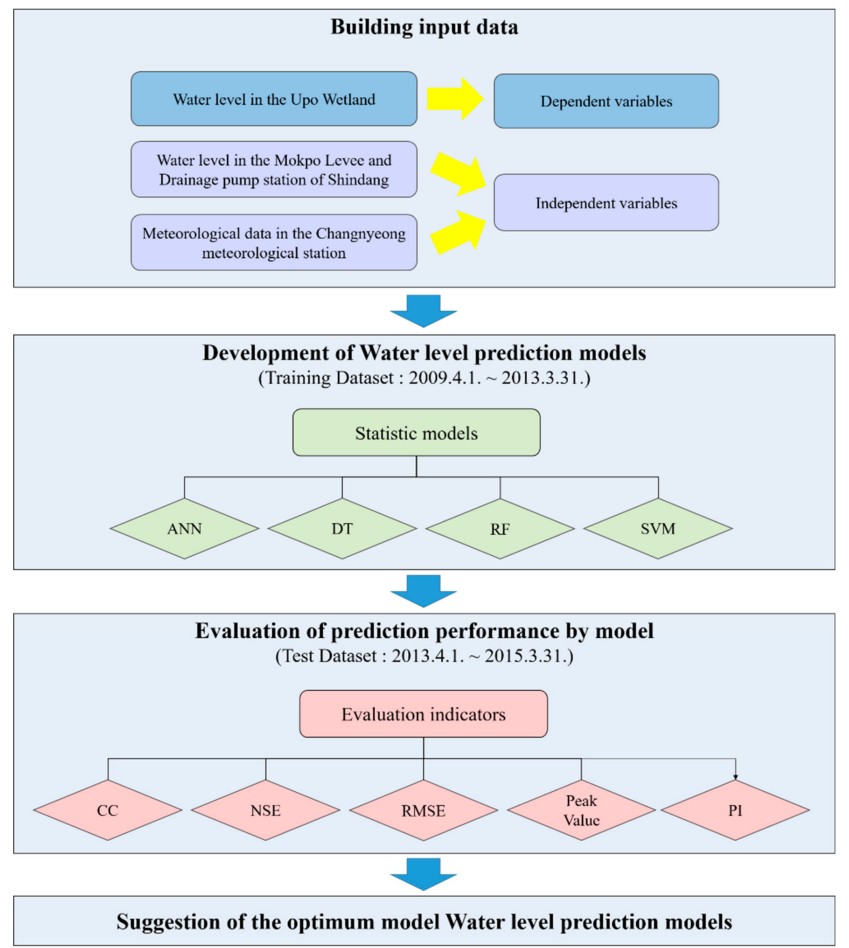

**Figure 1.** Flow chart of this study.

## 2. Materials and Methods

### 2.1. Study Area

The Upo wetland is the largest inland wetland located in Changnyeong-gun, Gyeongsangnam-do, Korea. It consists of four wetlands, i.e., Upo, Mokpo, Sajipo, and JjokJibeol wetlands (Figure 2). Upo, home to a wide variety of living organisms and biological species, covers the largest area. The Upo wetland is located in Topyeongcheon, the first tributary of the Nakdong River. Topyeongcheon is a local river located on the left bank, about 108 km from the Nakdong River estuary, with a basin area of 123 km² and a river extension of 30 km. In the middle and lower stream of Topyeongcheon where the Upo wetland is located, the watershed becomes geographically unclear, bordering the Hyeonchangcheon, a local stream to the north, and Changnyeongcheon to the south. Moreover, it is connected to the Nakdong River, which crosses the wetland in a south to west direction.

Changnyeong-gun, where the Upo wetland is located, has an annual average temperature of 13.6 °C and an average annual precipitation of 1231.5 mm [41]. The Upo wetland was designated as a Ramsar Wetland on 2 March 1998 and is representative of high-value biodiversity inland wetlands. It is home to more than 480 species of plants, 62 species of birds, 28 species of fish, 55 species of aquatic insects, and 12 species of mammals. Therefore, it is necessary to secure data on water levels that have a significant effect on the function of the Upo wetland so that it can be continuously protected and managed [42].

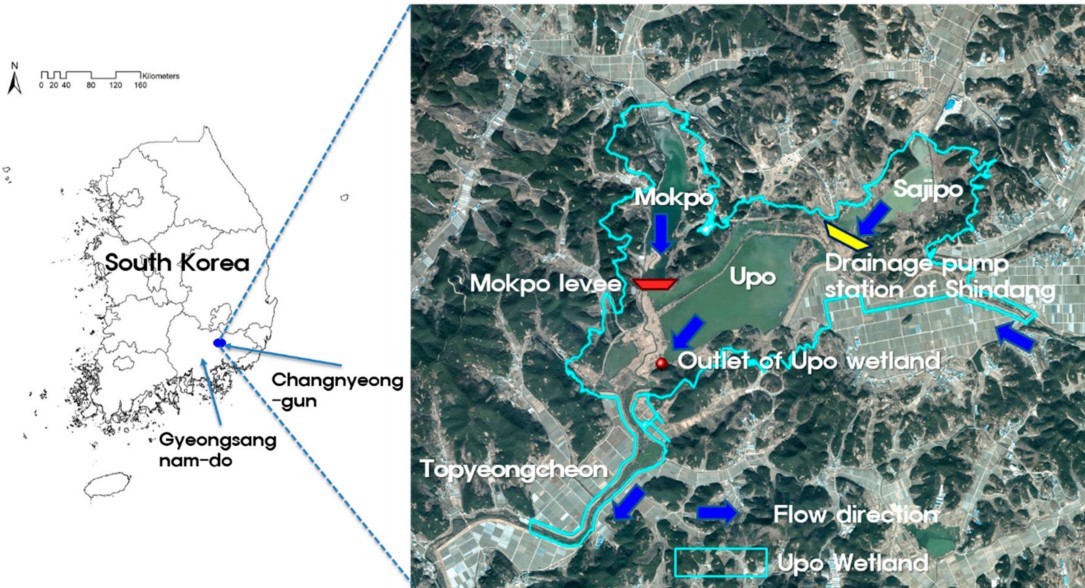

**Figure 2.** Study area (Upo wetland) of this study.

*2.2. Data Used*

2.2.1. Dependent Variables

To develop prediction models for predicting the water level of the Upo wetland, this study used the Upo wetland water level observation dataset from 1 April 2009 to 31 March 2015 as the dependent variable. The data had been measured by water level (depth of water) measuring equipment that had been installed and operated in the Upo wetland since 2009 for the efficient use and management of the Upo wetland. In this study, Orpheus mini, an automatic water level measuring device developed by the OTT company of Germany was used. In 2009 and 2010, the water levels were measured every 10 min and between 2011 and 2015, the water levels were measured every 15 min to improve storage memory efficiency. In this study, the water level data measured in 10-min or 15-min intervals were converted into daily average water level data and used as dependent variables.

Figure 3 shows the dependent variable over time. For the most part, the water level was maintained at 2–3 m. During the rainy season (June–August), there was a sudden change in the water level due to the heavy rainfall. In January 2011, the water level dropped drastically, and it is believed that this temporary drop in the water level was caused by construction work near the water level observation equipment. Using machine learning methods, the training period used in this study was from 1 April 2009 to 31 March 2013 and the testing period was from 1 April 2013 to 31 March 2015. Table 1 shows the distribution of data by training and test period. The water level below 3 m occupies 91–93% of the distribution and the 3–4 m has 6–7% of the distribution. High water levels of over 4 m occupies about 2%. The distribution of water levels in the training and test period is similar.

**Table 1.** Distribution of water level by period.

| Water Level | Training Period | Test Period |
|:-----------:|:---------------:|:-----------:|
| <3 m        | 90.91%          | 92.54%      |
| 3–4 m       | 7.04%           | 5.94%       |
| >4 m        | 2.05%           | 1.52%       |

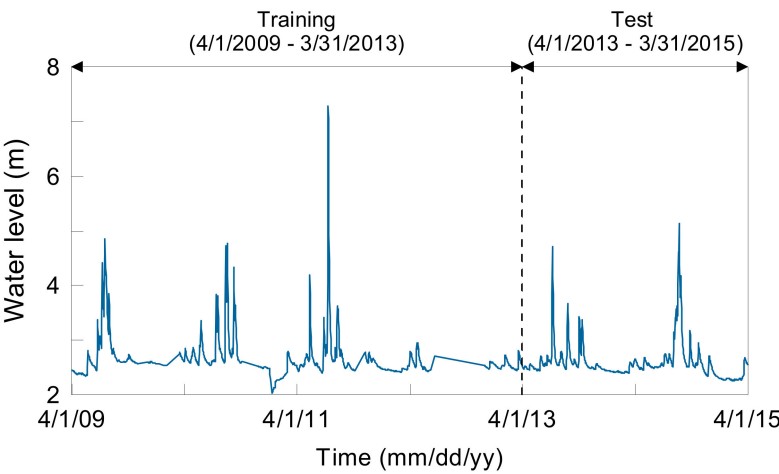

**Figure 3.** Time series of water level of the Upo wetland over time. Training period is from 1 April 2009 to 31 March 2013 and test period is from 1 April 2013 to 31 March 2015.

### 2.2.2. Independent Variables

As independent variables, the daily average temperature, daily minimum temperature, daily maximum temperature, daily precipitation, daily maximum instantaneous wind speed, daily average wind speed of Changnyeong station (approximately 6 km from the Upo wetland), and the water level data measured at the Mokpo embankment and Shindang drainage pump station situated upstream of the Upo wetland were used. Table 2 shows a list of independent variables. As our goal was to develop a model for predicting the future water level, we tried to predict the water level of the Upo wetland by using the weather and water level data obtained one to three days prior to the day of interest, as the independent variables.

**Table 2.** List of independent variables used in this study.

| Variable | Description | Variable | Description | Variable | Description |
|---|---|---|---|---|---|
| X1.1 | Average temperature (1 day ago) | X1.2 | Average temperature (2 days ago) | X1.3 | Average temperature (3 days ago) |
| X2.1 | minimum temperature (1 day ago) | X2.2 | minimum temperature (2 days ago) | X2.3 | minimum temperature (3 days ago) |
| X3.1 | Maximum temperature (1 day ago) | X3.2 | Maximum temperature (2 days ago) | X3.3 | Maximum temperature (3 days ago) |
| X4.1 | Precipitation (1 day ago) | X4.2 | Precipitation (2 days ago) | X4.3 | Precipitation (3 days ago) |
| X5.1 | Maximum instantaneous wind speed (1 day ago) | X5.2 | Maximum instantaneous wind speed (2 days ago) | X5.3 | Maximum instantaneous wind speed (3 days ago) |
| X6.1 | Average wind speed (1 day ago) | X6.2 | Average wind speed (2 days ago) | X6.3 | Average wind speed (3 days ago) |
| Z1.1 | Water level of Shindang (1 day ago) | Z1.2 | Water level of Shindang (2 days ago) | Z1.3 | Water level of Shindang (3 days ago) |
| Z2.1 | Water level of Mokpo (1 day ago) | Z2.2 | Water level of Mokpo (2 days ago) | Z2.3 | Water level of Mokpo (3 days ago) |

Figure 4 shows the mutual information of the independent and dependent variables. Mutual information (MI) is a statistic that can measure the degree of relatedness between datasets [43,44]. The mutual information between two random variables X and Y is defined in terms of their joint probability distribution $p(x, y)$ as shown in Equation (1):

$$MI(Y; X) = \sum_{x \in X} \sum_{y \in Y} p(x, y) \log\left(\frac{p(x, y)}{p(x) - p(y)}\right)$$

(1)

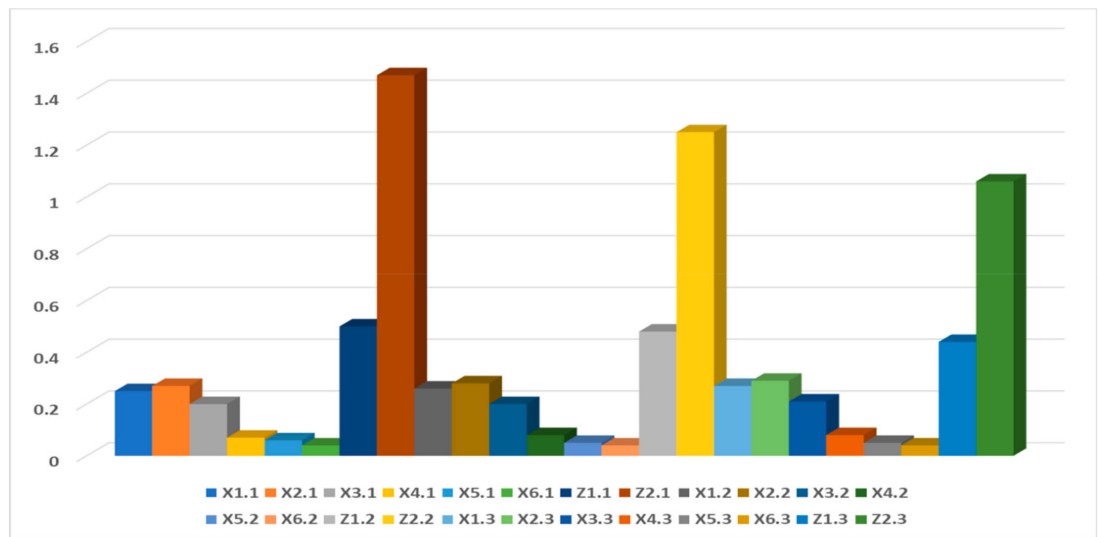

**Figure 4.** Mutual information by independent variables.

MI will be greater than zero when X and Y exhibit any mutual dependence (relatedness), regardless of how nonlinear that dependence is. The stronger the relatedness, the larger the value of the MI [45]. In this study, since there are many variables that are not linear in relationships, the correlation between the dependent and independent variables was examined using MI rather than the commonly used Pearson correlation (see Figure 5 for the scatter plots of the independent and dependent variables). The temperature (average, minimum, maximum) was found to be moderately related to the water level of the Upo wetland. The precipitation and wind speed (maximum instantaneous, average) exhibited a weak relatedness, and the upstream water level (Mokpo, Shindang) showed a strong relatedness. Most of the data from one day ago was found to have a higher relatedness than the data from two or three days ago. In the case of precipitation, mutual information was weak, but it has a linear relationship as shown in Figure 5.

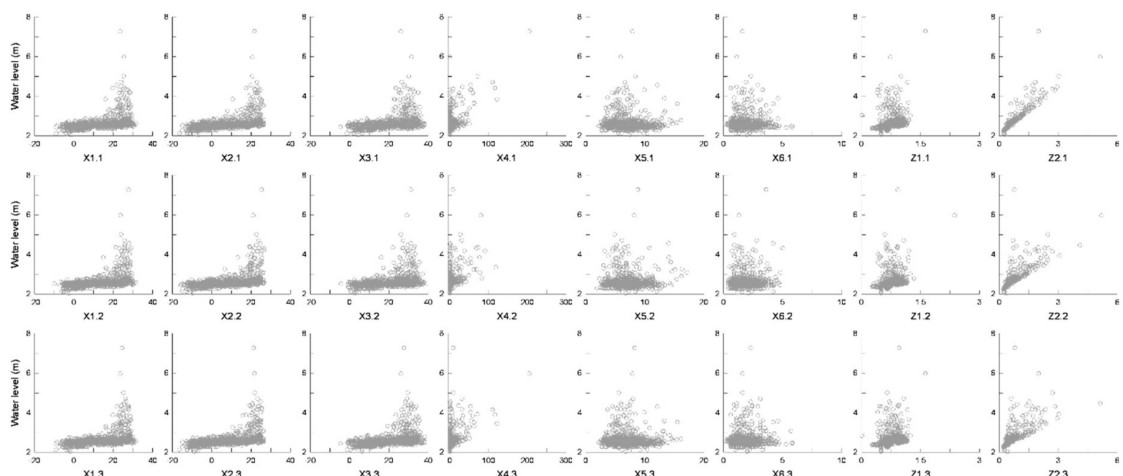

**Figure 5.** Scatter plots of independent and dependent variables.

*2.3. Machine Learning Techniques*

2.3.1. Overview

Machine learning can be applied to support the limitations of physical systems in which a large amount of data is operating under complex relationships [46]. In particular, the applicability of machine learning to predicting the process of continuous, nonlinear based hydrologic elements is very

high. Furthermore, machine learning has exhibited significant progress in forecasting and modeling non-linear hydrological applications [47]. Machine learning can be a very effective alternative for areas where the observation system is poor or where physical analysis is not possible, owing to the low density of the observation network. Therefore, this study predicted the water level by applying four machine learning methods (ANN, DT, RF, and SVM) to ungauged areas (e.g., wetland) where water level data is necessary, but available observed data is insufficient.

### 2.3.2. Artificial Neural Network

The artificial neural network (ANN) is one of the most widely used techniques in machine learning, and it refers to a computing system developed based on the neural network of the human brain [47]. ANNs are composed of neurons, which are units of basic computing, connected by several links. Here, each link may be weighted according to a given environment. The most common ANN model is a multilayer perceptron with multiple hidden layers between one input layer and one output layer. Depending on the structure and purpose of use for the model, various types of models can be constructed. Figure 6 shows the conceptual diagram of an ANN architecture. The neural network consists of three layers, and the number of nodes in each layer can be applied according to the environment. In other words, complex calculations are also possible, depending on the number of nodes in the hidden layer. However, the more complicated the structure, the computation process increases, causing over-fitting or generating a local minimum. Therefore, it is important to select an appropriate number of nodes according to the characteristics and conditions of the data. In addition, to run the neural network, it is essential to apply an activation function, which is a function for converting the sum of input signals into an output signal. In this study, the model was developed using the logistic activation function, which is selected from various activation function options.

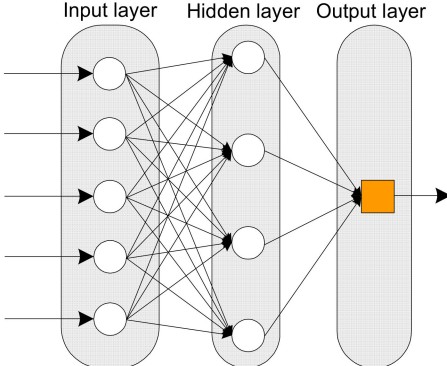

**Figure 6.** Conceptual diagram of an artificial neural network (ANN) model.

### 2.3.3. Decision Tree

The decision trees (DT), introduced by the authors of [27], is one of the machine learning techniques for classification and prediction, and it is used in various fields because of its easy implementation and high accuracy [48]. As the name suggests, a decision tree consists of a starting node (root) and a final terminal node (leaf) in the form of a tree (Figure 7). The DT classifies and predicts labels by substituting observation values of features in the model based on an analysis of the relationship between descriptive variables (features) and target variables (labels) for continuous data. The DT determines each variable and division values by using impurity generated at each node. It has the advantage of being very simple and easy to run; however, its reliability is low because it is susceptible to overfitting and a lack of linearity. Therefore, to minimize the occurrence of overfitting and error, it is necessary to determine the size of the tree using cross-validation.

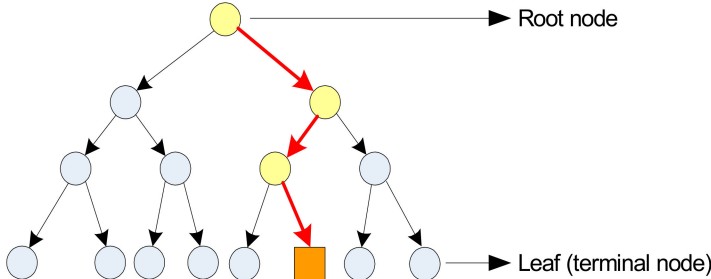

**Figure 7.** Conceptual diagram of decision tree (DT) model.

### 2.3.4. Random Forest

The random forest (RF) is a representative supervised learning technique aimed at classification and regression, and it is an ensemble learning method for optimal decision making based on the results of multiple decision trees [28]. The advantage of RF is that the calculation speed is fast, and the accuracy of the predicted results is high. The RF presents the final decision based on a combination of the results estimated from multiple decision trees, leading to high reliability of the obtained results and high model stability [26,49].

Figure 8 shows a conceptual diagram of a RF. The operating process of this model is as follows: (1) Randomly select n sub training sets from the total dataset. Here, the sub training set refers to a single decision tree; (2) a prediction value is extracted for each subset by running each sub training set (T1, T2, . . . , Tn) through the existing DT running method; and (3) extract the final outcome from the mean value of the results from each subset.

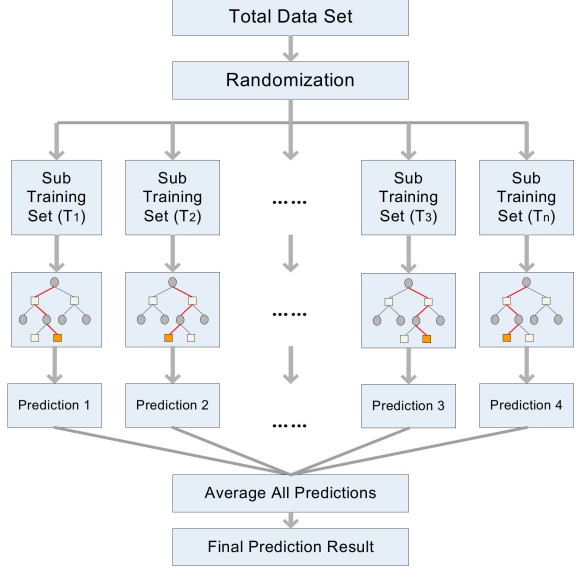

**Figure 8.** Conceptual diagram of random forest (RF) model.

### 2.3.5. Support Vector Machine

The support vector machine (SVM) can be applied to various types of data with very accurate classification results, after being presented by the authors of [29]. The SVM is a method of finding hyperplanes of support vectors that can linearly classify vectors of different classes with maximum margin for the distance between them. Figure 9 shows the conceptual diagram of the SVM.

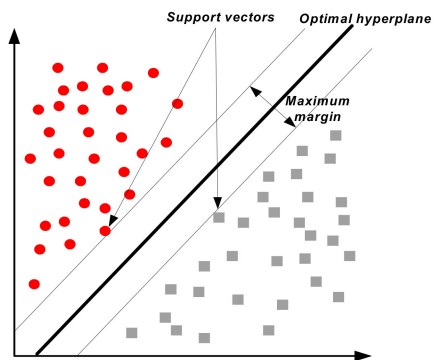

**Figure 9.** Conceptual diagram of the support vector machine (SVM).

The representative kernel functions are polynomial, sigmoid, and radial basis function. The SVM is mainly used to predict classification problems, and the extension of the SVM by means of introducing an $\varepsilon$-insensitive loss function to SVM to enable its use in the regression analysis is called support vector regression (SVR) [50,51]. In other words, the SVM is used to divide data into "+1" class and "−1" class in a classification problem, but the SVR is a generalized method of the SVM that is used to predict random real values [52,53]. Therefore, the SVR was used to predict the damage caused by heavy rain, and RBF, which is known to have a relatively high performance, was used as the kernel function. In this study, we developed a rainfall damage prediction model using an SVR with an "e1071" library of the R studio.

*2.4. Metrics for Evaluation*

In this study, three statistical indicators, correlation coefficient (CC), Nash–Sutcliffe efficiency (NSE), and root mean square error (RMSE), and peak water level, were used to evaluate the performance of the machine learning models applied to predict the water level at the ungauged areas. The peak values were compared to observation values. Three indicators were applied because they have different meanings, and the simulating performance of the characteristics of the water level (e.g., trend, peak value, error) for each model is different. CC is an indicator showing the degree of linear relationship between the simulated and observed data, the range of which is represented as −1 to 1. In other words, the closer the value of CC to 1, the stronger the positive correlation between the two datasets, while the closer the value of CC to −1, the stronger the negative correlation relationship. The NSE is an indicator that shows how well the plot of simulated and observed data fits on a 1:1 line; it is determined by the relative magnitude of the variance of the difference between the observed and simulated values (e.g., residual), and the variance of the observed data. NSE values between 0 and 1 indicate that the simulation values are generally acceptable, whereas negative NSE values indicate that the performance of the simulation value is very poor [54]. The RMSE, which is the standard deviation of the residual, namely the difference between the simulated and observed values, is used as an indicator of how much error the simulated results contain, as compared to the observed values. In addition, this study applied the persistence index (PI), which is one of the most commonly used benchmark methods to evaluate the phase and amplitude based error in the simulated water level from the model. The PI measures the relative magnitude of the residual variance to the variance of the errors estimated from the model. The optimal value of the PI is 1 and it should be larger than 0 to show minimally acceptable performance [55]. The equations for these statistics are as follows:

$$CC = \frac{\sum\left(Y_{sim} - \overline{Y_{sim}}\right)\left(Y_{obs} - \overline{Y_{obs}}\right)}{\sqrt{\sum\left(\left(Y_{sim} - \overline{Y_{sim}}\right)^2\right.}\sqrt{\sum\left(Y_{obs} - \overline{Y_{obs}}\right)^2}} \tag{2}$$

$$NSE = 1 - \frac{\sum(Y_{sim} - Y_{obs})^2}{\sum(Y_{obs} - \overline{Y_{obs}})^2} \tag{3}$$

$$RMSE = \sqrt{\frac{\sum(Y_{sim} - Y_{obs})^2}{n}} \tag{4}$$

$$PI = 1 - \frac{\sum(Y_{obs,i} - Y_{sim,i})^2}{\sum(Y_{obs,i} - Y_{obs,i-1})^2} \tag{5}$$

where, $Y_{obs}$ is the observed water level and $Y_{sim}$ is the modeled value from the model. $\overline{Y_{obs}}$, $\overline{Y_{sim}}$ are the average values of $Y_{obs}$ and $Y_{sim}$.

## 3. Results

### 3.1. ANN

For the ANN, the predictive performance is good if the values ranges between 0 and 1 [56]; therefore, the dependent and independent variables are standardized to values between 0 and 1 using the re-scaling method as shown in Equation (6). Instead of the conventional backpropagation algorithm, we used the resilient backpropagation with backtracking, a more robust version of the algorithm, and the logistic activation function [57,58]. The error function uses sum of the squared errors, which is popularly used [59,60]. In neural networks, the result changes according to the number of nodes. Although, in general, various complex features and abstracted properties can be extracted as the number of nodes increases making the neural network more complex. If the configuration is too complex, it tends to fall into the local minimum or greatly increase the computational process. Therefore, in this study, the optimal number of nodes was found by adjusting the number of nodes to 1–10, and retraining each node 100 times to determine the mean and standard deviation. Table 3 shows the RMSE results repeated 100 times for each node. Except in the case when there is only one node, then the RMSE was 0.15–0.16 and the standard deviation was 0.02–0.03, indicating a low variability.

$$Re-scaling = \frac{X_i - \min(X)}{\max(X) - \min(X)} \tag{6}$$

**Table 3.** Root mean square error (RMSE) of ANN by nodes.

| Node | Average | Standard Deviation | Node | Average | Standard Deviation |
|------|---------|--------------------|------|---------|--------------------|
| 1 | 0.19 | 0.08 | 6 | 0.15 | 0.03 |
| 2 | 0.15 | 0.05 | 7 | 0.16 | 0.03 |
| 3 | 0.15 | 0.03 | 8 | 0.16 | 0.02 |
| 4 | 0.15 | 0.03 | 9 | 0.16 | 0.02 |
| 5 | 0.15 | 0.02 | 10 | 0.16 | 0.03 |

### 3.2. Decision Tree

Decision trees generate trees by repeatedly dividing the data by finding branch points that minimize the mean squared error. As the process of dividing the data is repeated, the probability of overfitting increases, so cross-validation should be used to determine the tree size that minimizes the error. In this study, the average error was calculated for each cost complexity parameter (CP) representing the tree size through 10-fold cross validation. The pruning was carried out by finding the CP value with the lowest mean error, and the optimal DT model was thus determined. As shown in Figure 10a, the average error was calculated for each CP representing the size of the tree through 10-fold cross-validation. The size of the tree was determined by finding the CP value with the lowest error, and the optimal

function was developed by pruning. The result of the optimal DT model is shown in Figure 10b, and the water (Mokpo) level data from a day ago and the daily precipitation data from a day ago were found to have a significant effect. In case of a rapid increase in water level, the maximum instantaneous wind speed of three days ago was found to have a significant effect.

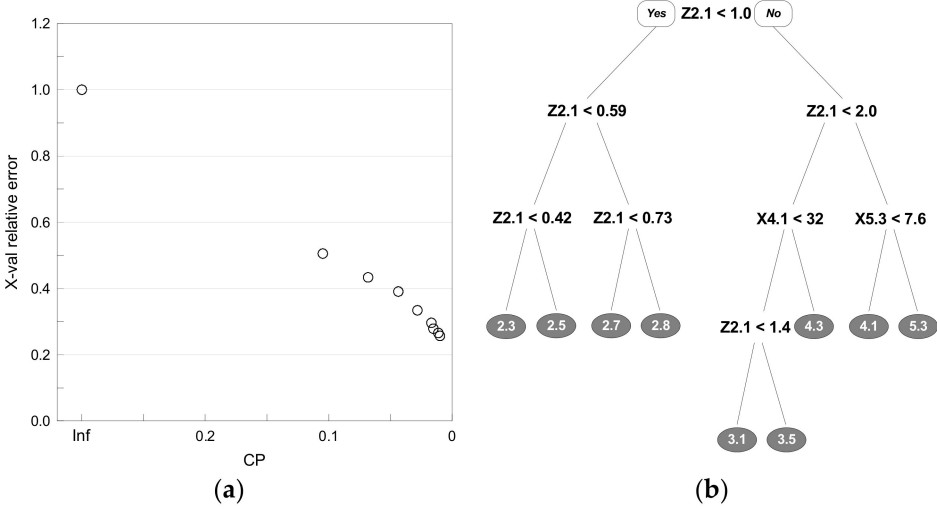

**Figure 10.** (**a**) Relative error by complexity parameter; (**b**) Level prediction model using decision tree.

### 3.3. Random Forest

Random forest consists of trees formed by selecting m/3 variables randomly from each split if the number of independent variables is m while the dependent forest is continuous, and typically 500 different trees are produced to derive results. However, if the number of trees is too large, there is a high risk of overfitting. Therefore, the optimal random forest model was determined by finding the number of trees that minimize the mean square error. As there are 24 independent variables, we randomly selected 8 variables (24/3) and generated 500 different trees to develop a function using a random forest. To optimize the model, the number of trees with the minimum error was confirmed, as shown in Figure 11a. As the error is minimized when the number of trees is 492, we randomly selected 8 variables and generated 492 different trees to derive the water level prediction model using the optimal random forest. According to Figure 11b, which shows the importance of each variable through the degree of increase in the node purity (IncNodePurity), the most important variables in predicting the water level of the Upo wetland were identified as the water level (Mokpo) of one, two days ago, the precipitation of one day ago, and the water level (Mokpo) of three days ago.

### 3.4. Support Vector Machine

The SVM was converted to the SVR to predict the water level, which is a regression problem, and the kernel function uses the radial basis function (RBF), which is known to have a relatively high accuracy for regression problems (in fact, in this study, the predictive power was evaluated using both polynomial and sigmoid functions, but only the results for the RBF were presented because their performances were lower than that of the RBF). The SVM selects an optimal parameter by adjusting the values of cost and the $\epsilon$-insensitive loss function, which determine the generalization of the regression function. In this study, 10-fold cross-validation was used following the previous studies to select the optimal parameters by adjusting the cost range to $2^{(0–7)}$ and the range of $\epsilon$ at 0.1 intervals from 0 to 1. The parameter condition to minimize the error was found to be cost = 8 and $\epsilon$ = 0.2.

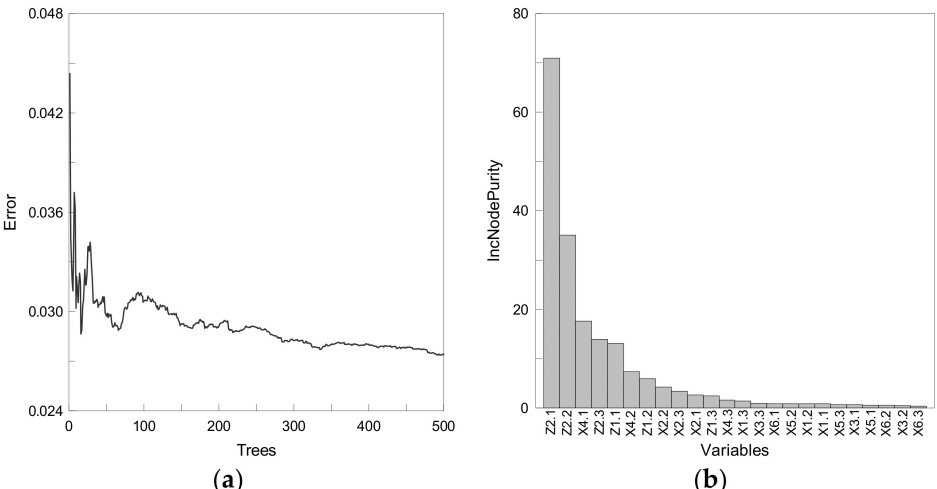

(**a**)　　　　　　　　　　　　　　　　　(**b**)

**Figure 11.** (**a**) Error by number of trees; (**b**) Importance of variables.

### 3.5. Evaluation Results

This study assessed the simulated water level from each machine learning models during the test period to evaluate the performance of models for water level prediction. Figure 12 shows the evaluation result. First, the average CC value was 0.94, which indicates that most of the machine learning models simulates the trends of observed water levels appropriately. The average value of the NSE was 0.82, indicating that the overall simulation results simulate the overall performance of the observed water level well. In addition, the mean value of the RMSE was 0.14. This indicates that the error of the simulated value is lower than the observed value.

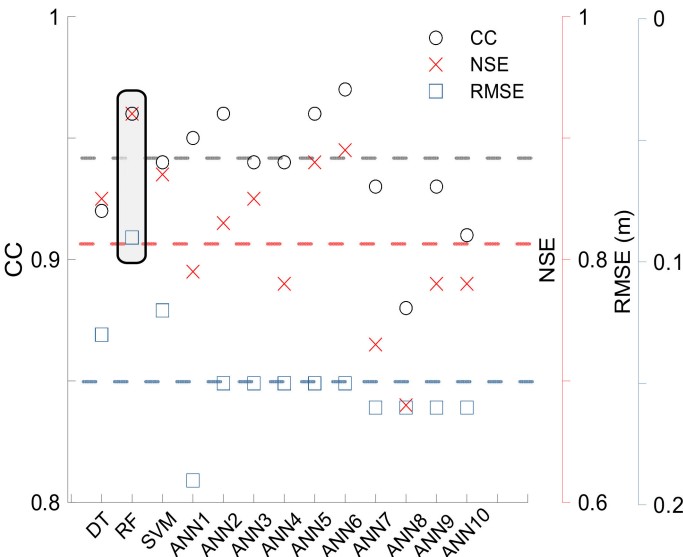

**Figure 12.** Correlation coefficient (CC), Nash–Sutcliffe efficiency (NSE), and RMSE values for machine learning models. The dotted lines indicate the average values of each index. The gray rectangles refer to the models showing the best performance.

These results indicate that the performance of machine learning models for water level prediction is reasonable and acceptable. Considering the results of the error indices for each model shown in Figure 12, it can be seen that the RF provide the best simulation performance. Results show that the CC and NSE are closest to the ideal value (e.g., 1.0) and have the lowest RMSE. Among the ANNs, six nodes performed the best. Table 4 shows the PI values computed on machine learning models.

The PI value is positive only for RF and negative for all the other models. Overall, the RF was the best method for simulating the water level among the machine learning models used in this study.

**Table 4.** Persistence index computed on machine learning models.

| Model | PI | Model | PI |
|-------|------|-------|------|
| DT | −0.62 | ANN5 | −0.25 |
| **RF** | **0.19** | ANN6 | −0.18 |
| SVM | −0.40 | ANN7 | −1.85 |
| ANN1 | −1.21 | ANN8 | −2.45 |
| ANN2 | −0.84 | ANN9 | −1.33 |
| ANN3 | −0.63 | ANN10 | −1.37 |
| ANN4 | −1.32 | | |

## 4. Discussion

In this study, the evaluation results using the error index and the differences in the simulated peak water levels from the models were evaluated. Figure 13 shows four peak water levels selected for the evaluation of peak levels. The peak level occurred in July, August, October 2013, and August 2014, and the values were 4.7 m, 3.7 m, 3.4 m, and 5.0 m. Table 5 shows the comparison of the peak water level and the time delay between the observed and simulated results of the models. The results reveal that RF exhibited good performance, and in the case of the peak level of October, 2013, the RF exhibited only a 0.6% difference. In terms of peak time delay error, RF presents the best performance with one-day delay error in four cases.

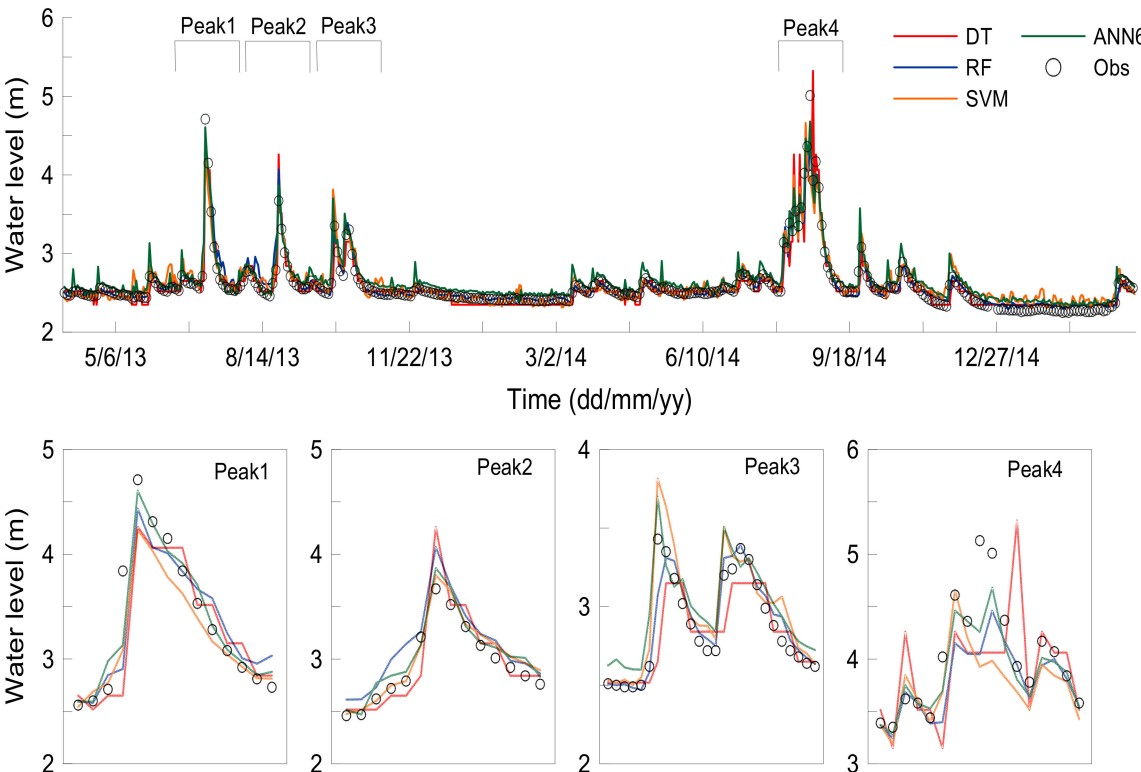

**Figure 13.** Time series of simulated and observed water level for test period (from 1 April 2013 to 31 March 2015). Four peak values (6 July 2013, 25 August 2013, 11 October 2013 and 22 August 2014) were selected for peak level evaluation.

**Table 5.** Differences between peak levels derived from the four machine learning (ML) techniques and observation.

| No | Date | ANN Peak (%) | ANN Time (day) | DT Peak (%) | DT Time (day) | RF Peak (%) | RF Time (day) | SVM Peak (%) | SVM Time (day) |
|----|------|----------|-----------|---------|----------|---------|----------|----------|-----------|
| 1 | 6 July 2013 | −2.2 | 0 | −9.6 | 0 | −5.8 | 0 | −10.1 | 0 |
| 2 | 15 August 2013 | 5.4 | 0 | 16.1 | 0 | 10.9 | 0 | 3.6 | 0 |
| 3 | 11 October 2013 | −3.4 | −2 | −6.5 | 0 | 0.6 | 0 | −2.5 | −2 |
| 4 | 22 August 2014 | −6.6 | 1 | −18.9 | 3 | −10.9 | 1 | −20.5 | −2 |

Figure 14 shows the scatter plot of the observed and predicted values for each model. ANN, as verified by previous studies, exhibited good performance. However, it had difficulties in predicting peak values. Its predictive performance also appears to be poor in comparison to RF. In the case of DT, the water level of the Upo wetland was predicted using the water level of one day ago (Mokpo), the rainfall of one day ago, and the maximum instantaneous wind speed of three days ago. This method showed greater errors compared to the other models. As the leaf of the DT was presented with a total of nine water level values (2.3, 2.5, 2.7, 2.8, 3.1, 3.5, 4.1, 4.3, 5.3), it showed that it could not predict the complex water level values well. The RF exhibited the best predictive performance and predicted most of the observation values except for some of the peak values. The importance of the water level (Mokpo) of one day ago, water level (Mokpo) of two days ago, the precipitation of one day ago, the water level (Mokpo) of three days ago and the average temperature of one day ago was also demonstrated. It was found that for the Mokpo embankment, which is the closest upstream area, the precipitation of one day ago and the average temperature of one day ago were important variables in predicting the water level of the Upo wetland. Thus, RF has the advantage of being able to identify important variables. In the case of SVM, relatively small values were well predicted, but peak values were not.

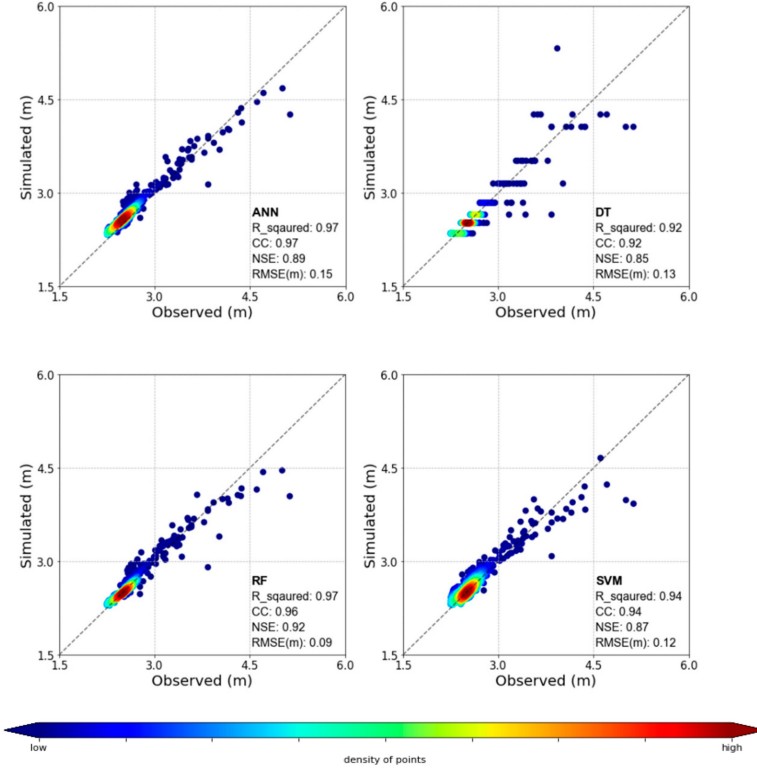

**Figure 14.** Comparison of the predicted water level with observation.

Based on the predictive performance evaluation results, the RF was the most suitable for simulating the water level among the ML techniques applied in this study. Therefore, in this study, the water level predictive model using the RF constructed by generating 492 different trees was selected as the water level predictive model of the Upo wetland.

## 5. Conclusions

The selected final RF model was shown to reflect the change in water level well. However, the reflection of the peak value was found to be somewhat insufficient. Therefore, further research is needed to address this problem. Due to the backflow phenomenon downstream of Nakdong River during the rainy season, flow rate returns to the Upo wetland. However, this study did not use backflow as an independent variable, leading to a decrease in the predictive performance. Therefore, it may be helpful for future studies to add independent variables (e.g., water level observation data at the downstream part of Upo wetland) which could reflect the backflow phenomenon during the rainy season. In addition, this study could not reflect the groundwater and soil types around the Upo wetland because of the limited data available. Thus, if the groundwater and soil properties in the Upo wetland is known, it will be very helpful in predicting the water level in the area. Although several variables are used in this study, choosing variables using mutual information or solving temporal/phase errors using error-weights will help to improve predictive performance.

Water level data, which plays a central role in the function of a wetland, has high applicability, but the observations have been limited owing to topographic reasons. Most of the previous studies predicted the water level using a single variable rather than multiple variables. Furthermore, the performance of many different machine learning techniques for water level prediction could not be confirmed, because previous studies used a single model such as ANN, which is the most widely used machine learning method. In this study, water level prediction models using machine learning was developed to predict the water level of the wetland. This model analyzed the correlation between wetland water level and meteorological data.

Based on the relationships between the water level and variables such as precipitation, temperature, and humidity, a predicted water level was obtained. We also comparatively analyzed the performance of machine learning techniques for water level prediction using various machine learning techniques (e.g., ANN, DF, RF, and SVM) to confirm the applicability of various machine learning techniques for water level prediction of a wetland.

Predictive performance evaluation of the water level prediction model using the selected RF revealed a CC value of 0.96, an NSE of 0.92, and an RMSE of 0.09. Apart from the peak values arising from the backflow in the downstream of Nakdong River during the rainy season, the model showed a good prediction of the Upo wetland water level. In particular, compared to the ANN used in the previous study, the RF model showed better predictive performance. The water level data results of this study can be used as basic data for the development of wetland management techniques which were found to be insufficient or unavailable before.

**Author Contributions:** Conceptualization, C.C., J.K. and H.H.; methodology, C.C. and H.H.; software, C.C. and H.H.; validation, C.C. and H.H.; formal analysis, C.C.; investigation, J.K.; resources, J.K.; data curation, C.C. and D.H.; writing—original draft preparation, C.C., J.K. and, H.H.; writing—review and editing, H.S.K.; visualization, H.H.; supervision, H.S.K.; project administration, H.S.K.; funding acquisition, H.S.K. All authors have read and agreed to the published version of the manuscript.

**Funding:** This work was supported by the National Research Foundation of Korea (NRF) grant funded by the Korea government (MSIT) (No. 2017R1A2B3005695).

**Conflicts of Interest:** The authors declare no conflict of interest.

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
