# Peer review of "Development of Water Level Prediction Models Using Machine Learning in Wetlands: A Case Study of Upo Wetland in South Korea"

_water, doi:10.3390/w12010093_

Round 1
Reviewer 1 Report
This manuscript presents the comparison of four ML techniques to predict the water level in a wetland in South Korea. The subject matter is appropriate for the journal Water, and this application would be of interest to many researchers globally. However, the manuscript has serious limitations, described below, which makes it currently unacceptable for publication. The major concern stems from the fact that the use of the ML models is not particularly novel (standard plug-and-play approach); no new insights are developed into either the methods or the case study; while forecasting water level in wetlands is less common - water level forecasting (in rivers) using the proposed approach (using time lagged hydrometeorological inputs) is widely researched and published (and the authors do not sufficiently review the previous work). Additionally:
the authors need to clearly describe the rational for selecting the four ML used in this research; it is not clear what the objective of such a comparison is. Some information is provided in Line 162-171 (which is very well-written), but this should be provided earlier in the manuscript. While the focus on wetland water levels is welcome, the statement ended on Line 40 must be supported by evidence (i.e. citations) The review of existing ML research, particularly ANN, for predicting water levels is extremely limited (line 44). The author's should review current state-of-the-art ANN and ML applications for water level prediction. They should they frame their research objectives in the context of what has already been published. In its current form, the existing review is far for complete or thorough. Line 65: the authors somewhat interchangeably use "statistical models" to refer to the ML models; this is incorrect and ML should be used consistently throughout the manuscript The research gaps are not well defined (see Line 70): the objective does not stem clearly from the identified gaps. The performance metrics used are not suitable: CC (linear), NSE, and RMSE arguably measure the same amplitude based error (see Gupta et al 2009, doi:10.1016/j.jhydrol.2009.08.003 for a complete discussion). It is important to include more suitable metrics that calculate phase and amplitude based error (see Persistence Index for a good example) Data-division: some analysis into whether both datasets for training and testing have similar properties should be described; the percentage of data used for each subset should be clearly defined Line 127: use linear correlation coefficient for hydrometeorological data is very problematic (and highlighted in Fig 4). Most of the data are not linearly correlated (we know this from the hydrological point of view); thus, using a linear correlation coefficient is not useful. I would encourage the authors to consider using Mutual Information, Partial mutual information, or even partial correlation as alternative means of determining the most useful inputs. Statement on Line 147-150: this is not very insightful without a description of the hydrological system point of view. Input variable selection (i.e., feature selection) should be used to remove redundant variables from the ML models: this will improve convergence, accuracy, and computation cost. This is especially important when a large number of lagged variables are used (As proposed in this research) The description of the ML methods is incomplete: full information on hyperparameter selection (and justification for doing so) should be provided. The language used should be much more specific, e.g., the logisitic activation function, I assume was only used for the hidden layer, not for the output layer. This is not clear. The stopping criteria (Line 275) does not seem appropriate - how is over fitting determined using such an approach? The methods should be described using the actual data used in the research - in its current form they read like sections of a textbook. This is not suitable for a research paper. Line 245: as mentioned above, using three amplitude based error metrics is not suitable for time series data where the interest is in both the amplitude and phase errors of the ML models. Note (as cited above), the three metrics do not have different meanings (refer to Gupta et al 2009) None of the symbols used in Eqns 1 to 3 are defined I commend the authors in using a trial-and-error approach to find the best ANN structure. However, a single trial is not sufficient. The authors must retrain each ANN x100 times to get an ensemble of results to determine the optimum model. This is significant since the random initialisation will have a large impact on the convergence of the model. Similarly, the difference between 5 and 6 nodes is minor. Furthermore, all analysis should be displayed for the TESTING subset only - no inference should be made using the TRAINING dataset. Model performance: Line 359: this timeseries plot is very important but is too small. The authors should show zoomed in versions of the timeseries and describe the obvious bias issues (as shown in Table 3), and delay in peak-prediction issues. Some of these limitations of the results can be addressed by improving the structure of the ML approaches used. Error-weights are one way to improve the temporal/phase error, as well as to address the issue of poor performance at high-water levels (Fig 15). Line 367: this is not correct - the inability to select the important independent variables is because only a linear correlation analysis was conducted - which obviously will not capture nonlinear relationships. A suitable non-linear method should be used to identify the important variables. Figure 2: quality is poor Figure 5: the subplots are too small (Text is not legible): linear fit lines are not necessary or helpful Figure 10 is not helpful: the values are illegibleAuthor Response
# Thank you for your careful review. The reviewer's comments helped to improve the quality of the paper. The authors tried to reflect the reviewer's opinion as much as possible.
This manuscript presents the comparison of four ML techniques to predict the water level in a wetland in South Korea. The subject matter is appropriate for the journal Water, and this application would be of interest to many researchers globally. However, the manuscript has serious limitations, described below, which makes it currently unacceptable for publication.
The major concern stems from the fact that the use of the ML models is not particularly novel (standard plug-and-play approach); no new insights are developed into either the methods or the case study; while forecasting water level in wetlands is less common - water level forecasting (in rivers) using the proposed approach (using time lagged hydrometeorological inputs) is widely researched and published (and the authors do not sufficiently review the previous work). Additionally: the authors need to clearly describe the rational for selecting the four ML used in this research; it is not clear what the objective of such a comparison is.
=> This study applies machine learning methods to develop the water level prediction model of the wetland which is relatively lacking in research. As what the reviewer has suggested, the purpose of this research and its difference from the other existing research has been added to the introduction.
Some information is provided in Line 162-171 (which is very well-written), but this should be provided earlier in the manuscript.
=> As what the reviewer has suggested, the authors moved the suggested paragraph (about the introduction to machine learning) to an earlier part in the manuscript (e.g. introduction section).
While the focus on wetland water levels is welcome, the statement ended on Line 40 must be supported by evidence (i.e. citations)
=> Line 38-41: The authors added a reference (See #9 reference).
The review of existing ML research, particularly ANN, for predicting water levels is extremely limited (line 44). The author's should review current state-of-the-art ANN and ML applications for water level prediction. They should they frame their research objectives in the context of what has already been published. In its current form, the existing review is far for complete or thorough.
=> Line 42-52: As what the reviewer has suggested, the authors reviewed the previous studies which predict the water level using ANN, then added some recent trends.
Line 65: the authors somewhat interchangeably use "statistical models" to refer to the ML models; this is incorrect and ML should be used consistently throughout the manuscript.
=> As what was recommended by the reviewer, the authors modified the words “statistical model” to “Machine learning(ML)”.
The research gaps are not well defined (see Line 70): the objective does not stem clearly from the identified gaps.
=> Line 63-75: The authors analyzed the recent trends in wetland water level predictions and clarified our objective in the study.
The performance metrics used are not suitable: CC (linear), NSE, and RMSE arguably measure the same amplitude based error (see Gupta et al 2009, doi:10.1016/j.jhydrol.2009.08.003 for a complete discussion). It is important to include more suitable metrics that calculate phase and amplitude based error (see Persistence Index for a good example)
=> As what was recommended by the reviewer, the authors applied the persistence index (PI) to evaluate the phase and amplitude based error estimated from the models (See Table 4.).
Data-division: some analysis into whether both datasets for training and testing have similar properties should be described; the percentage of data used for each subset should be clearly defined
=> As what was recommended by the reviewer, the authors added the percentage of data used for each subset (Table 1.).
Line 127: use linear correlation coefficient for hydrometeorological data is very problematic (and highlighted in Fig 4). Most of the data are not linearly correlated (we know this from the hydrological point of view); thus, using a linear correlation coefficient is not useful. I would encourage the authors to consider using Mutual Information, Partial mutual information, or even partial correlation as alternative means of determining the most useful inputs.
=> As what was recommended by the reviewer, the authors considered the use of Mutual Information (See Figure 4.).
Statement on Line 147-150: this is not very insightful without a description of the hydrological system point of view. Input variable selection (i.e., feature selection) should be used to remove redundant variables from the ML models: this will improve convergence, accuracy, and computation cost. This is especially important when a large number of lagged variables are used (As proposed in this research)
=> In the case of linear regression model, the use of multiple variables causes problems such as multicollinearity, and thus the predictive performance is often degraded. However, in the case of machine learning, such problems do not occur even if multiple variables are being used. Thus, using multiple variables is not an issue in machine learning. However, as what was suggested by the reviewer, selecting variables using MI might be considered. The authors suggest to leave it as part of the limitations of the research so that the authors can proceed later. The limitations were written in the conclusion section. Thank you for your good opinion.
The description of the ML methods is incomplete: full information on hyperparameter selection (and justification for doing so) should be provided. The language used should be much more specific, e.g., the logisitic activation function, I assume was only used for the hidden layer, not for the output layer. This is not clear. The stopping criteria (Line 275) does not seem appropriate - how is over fitting determined using such an approach? The methods should be described using the actual data used in the research - in its current form they read like sections of a textbook. This is not suitable for a research paper.
=> As what was suggested, the authors have written an additional explanation for ML.
Line 245: as mentioned above, using three amplitude based error metrics is not suitable for time series data where the interest is in both the amplitude and phase errors of the ML models. Note (as cited above), the three metrics do not have different meanings (refer to Gupta et al 2009) None of the symbols used in Eqns 1 to 3 are defined.
=> The authors applied the persistence index (PI) as well as the three metrics (CC, NSE and RMSE). The authors expect that it can evaluate the phase and amplitude based error of the models for predicting the water level. In addition, the authors defined the symbols of equations in the manuscript.
I commend the authors in using a trial-and-error approach to find the best ANN structure. However, a single trial is not sufficient. The authors must retrain each ANN x100 times to get an ensemble of results to determine the optimum model. This is significant since the random initialisation will have a large impact on the convergence of the model. Similarly, the difference between 5 and 6 nodes is minor. Furthermore, all analysis should be displayed for the TESTING subset only - no inference should be made using the TRAINING dataset.
=> Each node was performed 100 times, and the mean and standard deviation of the results were presented in a table. As what was suggested by the reviewer, we have divided the training dataset and the test dataset clearly.
Model performance: Line 359: this timeseries plot is very important but is too small. The authors should show zoomed in versions of the timeseries and describe the obvious bias issues (as shown in Table 3), and delay in peak-prediction issues. Some of these limitations of the results can be addressed by improving the structure of the ML approaches used. Error-weights are one way to improve the temporal/phase error, as well as to address the issue of poor performance at high-water levels (Fig 15).
=> As what was recommended by the reviewer, the authors revised this figure. Using the error-weight seems to be a good idea. the authors will leave this as a threshold so that we can proceed later. Thank you for good insight.
Line 367: this is not correct - the inability to select the important independent variables is because only a linear correlation analysis was conducted - which obviously will not capture nonlinear relationships. A suitable non-linear method should be used to identify the important variables.
=> Since the ANN model is a black box model, it is difficult to know what the important variables are. The authors added some descriptions and references for the content.
Figure 2: quality is poor
=> As what was recommended by the reviewer, the authors modified Figure 2 and then used Figure 2 as a reference to the location of the study area
Figure 5: the subplots are too small (Text is not legible): linear fit lines are not necessary or helpful
=> As what was recommended by the reviewer, the authors revised this figure.
Figure 10 is not helpful: the values are illegible
=> As you advised, the authors deleted Figure 10.
Reviewer 2 Report
A clear and well written paper comparing different machine learning algorithms for predicting water levels in wetlands with sparse historical data. Most of my comments (below) are editorial in nature (with suggestions for improvement). I have one overarching question for the authors which was not clear to me from the text: When comparing the ANN approach, why only focus on one hidden layer?
Comments:
Line 38: “… Ramsar Designated Wetlands… ” is mentioned but not where it is located (South Korea) or referenced by a figure in the text.
Line 72: (same as previous comment). Maybe using Figure 2 as a reference to the study area location.
Line 86: In Figure 1, the middle schematic uses the title “Statistical models”. While this is true in general, a more appropriate suggested title is “Machine Learning models”.
Line 89-90: the left-hand side of Figure 2 does not label the country (South Korea), nor included in the caption of the figure.
Line 91: Same as previous comment of using Figure 2 as reference for the wetlands area location.
Line 118: “…water level”. Is that depth of water or level to some reference (e.g., Mean Sea Level, etc.)?
Line 162: “… main theory...”. Suggest using “... main objective…”.
Line 164-165: “…dependent variable”. Machine learning uses one or more variables, including this paper. Suggest changing to “…dependent variables”.
Line 166: “...the decision tree”. Standard nomenclature is to reference the method as “Decision Trees” in the plural and not “decision tree” in the singular.
Line 172: “...to support for the limitations...”. Suggest changing to “…to support the limitations …”.
Line 189: ”…of an ANN.” Suggest rewording to “….of an ANN architecture.”
Lines 189-197: The paper focuses on one hidden layer only. However, the power of ANN is utilizing multiple hidden layers (also known as deep learning), with much fewer nodes per hidden layer than when considering one single hidden layer. There are other forms of ANN as well, including Recurrent Neural Networks, and Convolution Neural Networks, Long Short- Term Memory neural networks, etc.
Line 193: “…to select an appropriate node according …”. I think the authors meant “…to select an appropriate number of nodes according ….”.
Line 201: “The decision tree introduced…”. Suggest modifying to “Decision Trees introduced….”
Line 202: “… easy operation…” suggest changing to “… easy implementation…”
Line 235: SVR is mentioned but there is no reference in the “References”.
Lines 240-241: “Figure 9… diagram of the SVR”. Figure 9 shows SVM (caption).
Line 252: “… of CC is to 1 …”. True for positive correlation; negative correlation would be -1.
Line 272-273: “… used resilient backpropagation with backtracking, which …”. Method mentioned but not referenced in “References”.
Line 276-282: The authors are using only one hidden layer in the ANN architecture. It should be noted that more hidden layers would also provide a more robust technique.
Line 493-494: Reference #29 is included but not referenced to in the main text.
Author Response
# Thank you for your careful review. The reviewer's comments helped to improve the quality of the paper. The authors tried to reflect the reviewer's opinion as much as possible.
A clear and well written paper comparing different machine learning algorithms for predicting water levels in wetlands with sparse historical data. Most of my comments (below) are editorial in nature (with suggestions for improvement). I have one overarching question for the authors which was not clear to me from the text: When comparing the ANN approach, why only focus on one hidden layer?
=> An ANN with two or more hidden layers is called a DNN. In this study, only one hidden layer or ANN was used. DNN will be applied in future studies.
Comments:
Line 38: “… Ramsar Designated Wetlands… ” is mentioned but not where it is located (South Korea) or referenced by a figure in the text.
Line 72: (same as previous comment). Maybe using Figure 2 as a reference to the study area location.
Line 89-90: the left-hand side of Figure 2 does not label the country (South Korea), nor included in the caption of the figure.
Line 91: Same as previous comment of using Figure 2 as reference for the wetlands area location.
=> As what was recommended by the reviewer, the authors modified Figure 2 and then used Figure 2 as a reference to the location of the study area
Line 86: In Figure 1, the middle schematic uses the title “Statistical models”. While this is true in general, a more appropriate suggested title is “Machine Learning models”.
=> As what was recommended by the reviewer, the authors modified the words “statistical model” to “Machine learning(ML)”.
Line 118: “…water level”. Is that depth of water or level to some reference (e.g., Mean Sea Level, etc.)?
=> The part you pointed out means depth of water, and the authors added more contents so it will not make the readers/reviewers confused.
Line 162: “… main theory...”. Suggest using “... main objective…”.
=> As what was suggested by the reviewer, the authors revised the sentence.
Line 164-165: “…dependent variable”. Machine learning uses one or more variables, including this paper. Suggest changing to “…dependent variables”.
=> As what was suggested by the reviewer, the authors revised the sentence.
Line 166: “...the decision tree”. Standard nomenclature is to reference the method as “Decision Trees” in the plural and not “decision tree” in the singular.
=>
As what was suggested by the reviewer, the authors revised the sentence.
Line 172: “...to support for the limitations...”. Suggest changing to “…to support the limitations …”.
=> As what was suggested by the reviewer, the authors revised the sentence.
Line 189: ”…of an ANN.” Suggest rewording to “….of an ANN architecture.”
=> As what was suggested by the reviewer, the authors revised the sentence.
Lines 189-197: The paper focuses on one hidden layer only. However, the power of ANN is utilizing multiple hidden layers (also known as deep learning), with much fewer nodes per hidden layer than when considering one single hidden layer. There are other forms of ANN as well, including Recurrent Neural Networks, and Convolution Neural Networks, Long Short- Term Memory neural networks, etc.
Line 276-282: The authors are using only one hidden layer in the ANN architecture. It should be noted that more hidden layers would also provide a more robust technique.
=> In this study, we applied ANN using one hidden layer. We used ANN instead of DNN, due to the complicated computational process and long calculation time of the latter. Of course, we plan to develop a water level prediction model using deep learning such as MLP, LSTM, RNN and CNN and compare the performance with machine learning based model.
Line 193: “…to select an appropriate node according …”. I think the authors meant “…to select an appropriate number of nodes according ….”.
=> As what was suggested by the reviewer, the authors revised the sentence.
Line 201: “The decision tree introduced…”. Suggest modifying to “Decision Trees introduced….”
=> As what was suggested by the reviewer, the authors revised the sentence.
Line 202: “… easy operation…” suggest changing to “… easy implementation…”
=> As what was suggested by the reviewer, the authors revised the sentence.
Line 235: SVR is mentioned but there is no reference in the “References”.
=> As what was suggested by the reviewer, the authors added a reference for SVR.
Lines 240-241: “Figure 9… diagram of the SVR”. Figure 9 shows SVM (caption).
=> As what was suggested by the reviewer, the authors revised the sentence.
Line 252: “… of CC is to 1 …”. True for positive correlation; negative correlation would be -1.
=> As what was suggested by the reviewer, the authors revised the sentence.
Line 272-273: “… used resilient backpropagation with backtracking, which …”. Method mentioned but not referenced in “References”.
=> The authors added a reference to resilient backpropagation with backtracking
Line 493-494: Reference #29 is included but not referenced to in the main text.
=> Line 108: Reference #29 is changed to #42 and was referenced in the main text.
Round 2
Reviewer 1 Report
Thank you for making the extension revisions from my first review.